



# Lidar observations of Cirrus clouds at Palau island (7°33′ N, 134°48′ E)

Francesco Cairo[1], Mauro De Muro[1,2], Marcel Snels[1], Luca Di Liberto[1], Silvia Bucci[3], Bernard Legras[3], Ajil Kottayil[4], Andrea Scoccione[1,5], and Stefano Ghisu[6]

[1]Institute of Atmospheric and Climate Sciences, National Research Council of Italy (CNR), I-00133, Roma, Italy
[2]Now at: : AIT Thales Alenia Space, Roma, Italy
[3]Laboratoire de Météorologie Dynamique (LMD), UMR CNRS 8539, CNRS, IPSL, ENS-PSL, École Polytechnique, Sorbonne Université, Paris, France
[4]Advanced Centre for Atmospheric Radar Research, Cochin University of Science and Technology, Cochin, India
[5]Now at: Centro Operativo per la Meteorologia, Aeronautica Militare, Pomezia, Italy
[6]Università degli Studi di Roma "Tor Vergata", Dipartimento di Fisica, Roma, Italy

**Correspondence:** Francesco Cairo (f.cairo@isac.cnr.it)

**Abstract.** A polarization diversity elastic backscatter lidar has been deployed in the equatorial island of Palau in February and March 2016, in the framework of the EU Stratoclim project. The system operated unattended in the Atmospheric Observatory of Palau Island from 15 February to 25 March 2016, during nighttime. Each lidar profile extends from the ground to 30 km height. Here the dataset is presented and discussed in terms of the temperature structure of the UTLS, obtained from co-located radiosondings. During the campaign, several high altitude clouds were observed, peaking approximately 3 km below the Cold Point Tropopause (CPT) located above 17 km. Their occurrence was associated with cold anomalies in the Upper Troposphere (UT). Conversely, when warm UT anomalies occurred, the presence of cirrus was restricted to a 5 km thick layer centered 5 km below the CPT. Thin and subvisible cirrus were frequently detected close to the CPT. The particle depolarization ratios of these cirrus was generally lower than the values detected in the UT clouds. CPT cirrus occurrence showed a correlation with cold anomalies likely triggered by stratospheric wave activity penetrating the UT. The back trajectories study revealed a thermal and convective history compatible with the convective outflow formation for most of the cirrus clouds, suggesting that the majority of air masses related to the clouds had encountered convection in the past and had reached the minimum temperature during its transport in less than 48 hours before the observation. A subset of SVC, with low depolarization and high lidar ratio and with no sign of significative recent uplifting, may be originated in situ.

## 1 Introduction

Cirrus are clouds composed of ice particles which form in the upper troposphere, covering about 20–25 % of the Earth (Rossow and Schiffer, 1999). They play an important role in climate: cirrus control the amount of solar radiative energy reaching the



ground, reflecting a fraction of the incident sunlight back to space; they also absorb infrared radiation, thus modulating the loss
to space of the energy emitted from the surface and lower atmosphere. For their relevant role and feedbacks on climate, cirrus
have raised attention since long (Ramanathan and Collins, 1991). Moreover, cirrus are an essential modulator of the water
budget in the upper troposphere and in the stratosphere: by condensing water vapour and removing it by particle gravitational
settling, these clouds can dehydrate the upper layers of the troposphere and influence the amount of water vapor reaching the
stratosphere, a most important role especially in the Tropics where the penetration of tropospheric air into the Stratosphere
finds itself in the ascending branch of the Brewer-Dobson circulation, therefore impacting the Stratosphere as a whole. Their
radiative properties and dehydration potential depend critically on their microphysical properties, i.e. ice particles size, shape
and number density, which are determined by environmental conditions (water vapor abundance, temperature and dynamics of
the airmass where they reside) and formation mechanisms. Sassen and Cho (1992) suggest subdividing cirrus according to their
optical thickness $\tau$ into subvisible (SVC), thin and opaque ($\tau < 0.03$, $0.03 < \tau < 0.3$ and $\tau > 0.3$, respectively). Most of the thin
and SVC cirrus clouds occur in the Tropics and are more frequent at night and over the ocean. In the Tropics, cirrus often appear
to be associated with convective activity, with the likely exception of some of the highest ones in the Tropical Tropopause
Layer (TTL). The TTL is a transition region a few kilometers thick that separates the Troposphere and the Stratosphere and is
characterized by a radiatively driven slow ascent, and thus considered to be a processing region for tropospheric air to enter
the stratosphere (Fueglistaler et al., 2009). There, the presence of cirrus clouds is of particular interest as they both modify the
energy budget of the region through their radiative properties (Fu et al., 2018) and the release or uptake of latent heat upon their
formation or dissipation (Spichtinger, 2014). Moreover they can process tropospheric air both chemically and microphysically,
altering the water vapour budget of the layer (Flury et al., 2011) and eventually of the stratosphere as a whole.

There are basically two kind of processes responsible for the formation of cirrus, namely in situ and convective formation.
The first process is triggered by cooling from both large-scale vertical uplifting (Jensen et al., 1996; Pfister et al., 2001)
and atmospheric Kelvin or gravity wave activity (Immler et al., 2008; Fujiwara et al., 2009). Upon cooling, ice particles
may directly form by either homogeneous and/or heterogeneous nucleation. Krämer et al. (2015) suggests considering two
subclasses depending on the strength of the updraft and hence the rapidity of the cooling; slow updrafts tend to produce thinner
cirrus with lower Ice Water Content (IWC), while faster updrafts form thicker cirrus with higher IWCs. In the second process
cirrus are formed when deep convective mixed phase clouds deliver particles directly into the upper troposphere (Pfister et al.,
2001; Comstock and Jakob, 2004) where the temperature is low enough ($< 235$ K) to allow for the full freezing of the liquid
droplets. These convectively generated cirrus are generally thicker than those formed by in situ mechanisms.

Lidar techniques can detect thin and SVC cirrus with high spatial and temporal resolution, providing accurate information on
their geometrical and optical properties. Lidar backscattering ratio can be associated with cirrus microphysical bulk properties
(Cairo et al., 2011) and lidar depolarization and extinction-to-backscatter ratio (a.k.a. lidar ratio, LR), with the average particle
shape and phase, albeit with some caveats (Chen et al., 2002; Del Guasta, 2001; Reichardt et al., 2002).

The first report of lidar measurements of high tropical cirrus clouds, extending from 12 to 18 km with geometrical thickness
less than 1 km, are by Uthe et al. (1977) from Kwajalein island ($8°7'$ N, $116°7'$ E). Platt et al. (1987) compared tropical
(Darwin $12°8'$ S, $130°7'$E) and midlatitude (Aspendale $38°0'$ S, $144°0'$) cirrus using the LIRAD method and finding less





variability in depolarization and greater optical thickness at lower temperatures in tropical cirrus clouds; in later measurements

from Kavieng ($2°50'$ S, $152°7'$E) Platt et al. (1998) reported a decrease of the lidar integrated attenuated depolarization ratio with temperature, from 0.42 at -70°C to 0.18 at -10°C, which actually contradicts some later findings. For instance, Sassen and Benson (2001) show for midlatitude cirrus a quite regular increase for decreasing temperatures, from values around 0.3 at 240 K to 0.45 at 195 K; depolarization measurements from Mahe ($4.4°$ S, $55.3°$E) from Pace et al. (2003) do not show a regular behavior in temperature, while in observations from Gadanki ($13.5°$N, $79.2°$E) Sunilkumar and Parameswaran (2005)

observed an increase with decreasing temperature, albeit in both cases the depolarization dropped to its lowest values at the lowest temperatures (i.e. highest altitudes) observed. These apparently contradictory findings suggest that there may not be an univocal relationship between temperature and depolarization, as the latter may be influenced by the history rather than the instantaneous value of the air mass temperature, with high depolarization produced by fresh particles in cirrus clouds originating from the outflow of convective cells and intermediate to low depolarization associated with aged outflows and in

situ formed cirrus, the latter often in the form of SVC, dwelling at or slightly below the Tropopause, as suggested by Pace et al. (2003). Cirrus at the tropical tropopause are of particular interest as they represent the last thermodynamic process of water vapor before it enters the Stratosphere. Nee et al. (1998) recorded the presence of high tropospheric SVC, often showing an optical thickness of less than 0.01, at Chung-Li ($25°$ N, $121°$E), for almost 50% of the observational period between May and September 1993–95. Boehm and Verlinde (2000) related the occurrences of high and persistent cirrus clouds in Nauru ($0.5°$S,

$170°$E) to the temperature perturbation induced by equatorial Kelvin waves. The role of Kelvin as well as gravity- and Rossby waves and their link to extensive, persistent laminar cirrus has then been addressed extensively (Pfister et al., 2001; Garrett et al., 2004). Wang et al. (2019) shows from 10 years of lidar satellite observations that these optically thin laminar cirrus occurs frequently in the west/central tropical Pacific, equatorial western Africa, and northern South America, thus preferably in the tropical large-scale ascending zones. Similarly, Wang and Dessler (2012) have shown with satellite observations that

the convective fractions of cirrus increase with height until the cold point tropopause is reached, peaking in their geographical occurrence over equatorial Africa, the tropical western Pacific, and South America. They have reported at least ∼30 % of cirrus in the TTL are of convective origin. Some studies have focused on the western Pacific tropical warm pool (TWP) (Platt et al., 2002; Heymsfield et al., 1998; Sassen et al., 2000; Comstock et al., 2002). They have reported that there thin cirrus are particularly frequent and can cover more than 50 % of the area (Prabhakara et al., 1993; Dessler and Yang, 2003).

This area, which spans the western waters of the equatorial Pacific, is characterised by a mean sea surface temperature (SST) exceeding $28°$C, weak trade winds, deep convection extending above 15 km, and holds the warmest seawaters in the world. It has a large effect on surrounding monsoon regions and on climate so as to be called the "heat engine of the world" influencing remote regions and large scale weather variability (De Deckker, 2016). It is also a region of primary importance for the troposphere to stratosphere transport: with a Lagrangian trajectory analysis of tropospheric airmasses entering the stratosphere.

Kremser et al. (2009) have demonstrated that nearly half of the mass of air entering the stratosphere in NH winter, has reached its individual absolute temperature minimum during transport through the TTL in the TWP . Unfortunately, this area constitutes a gap in existing observational networks such as SHADOZ (Southern Hemisphere ADditional OZonesondes) or SOWER (Soundings of Ozone and Water in the Equatorial Region) and information on atmospheric composition from this region is very





limited. To improve this observational gap, in the framework of the EU funded project StratoClim, the Alfred Wegener Institute,

Helmholtz Centre for Polar and Marine Research (AWI) and the Institute for Environmental Physics of the University Bremen have set up a new ground station in the central West Pacific warm pool area, at Palau Island (7°N, 134°E). This Atmospheric Observatory operated in close collaboration with the Palau Community College during the timeframe of the project. The station there has been gradually equipped with a ground based solar absorption Fourier Transform Infrared Spectrometer (FTIR), balloons with ECC ozonesondes, water vapour sondes, backscatter sondes, a Max-DOAS Pandora instrument for

measuring trace gases in 2017 and in 2018 the COMCAL lidar for measurements of stratospheric and upper tropospheric aerosols (Immler et al., 2008). It hosted a small aerosol lidar from the Institute of Atmospheric Sciences and Climate of the Italian National Research Council in February and March 2016. The lidar operated during nighttime from 15 February to 25 March 2016, collecting atmospheric profiles of aerosol and clouds from the ground to 30 km. Cirrus clouds were observed on several occasions. We present here an analysis of the data obtained during that campaign. The instrument and the data

processing are described and the measurements are discussed, aiming at characterizing cirrus morphology, optical properties and connection with TTL temperature. A backtrajectory analysis is also presented, trying to connect the characteristics of the cirrus with their origin and mechanism of formation.

## 2  Instrument and data processing

### 2.1  The lidar system

The lidar deployed in Palau is a version of a small, portable, home-made instrument already described in Cairo et al. (2012). Similar systems are designed to work unattended in remote sites and have been used in previous campaigns in Africa (Cavalieri et al., 2010, 2011) and Europe (Di Liberto et al., 2012; Rosati et al., 2016; Bucci et al., 2018). We will briefly describe the setup used in this work. The system is contained in a 30 × 40 × 50 cm aluminum box, electronically shielded and thermally insulated with polyurethane. An inclined quartz window on its top allows the transmission of the laser beam and the collection

of the backscattered signal. The temperature in the aluminum box is controlled by four cooler-heater Peltier cells, 20 W each; an additional 200 W Peltier cell conditioner has been added to improve the temperature control in equatorial conditions. The laser source (Bright Solutions, Wedge) is an air cooled, diode pumped Nd-YAG, with second-harmonic generation and active Q switching. The laser pulse duration is 1 ns and the emission is at 532 nm (green) with energies of 1 mJ/pulse The pulse repetition rate is 1 kHz. The laser beam divergence is reduced to 0.4 mrad by a beam expander and the laser is aligned to the

telescope Field of View (FOV) by a steerable dielectric mirror, placed before the beam expander. The telescope is Newtonian with a diameter of 20 cm, f /1.5, with a FOV of 0.75 mrad. The overlap of the laser beam with the FOV begins at 40 m from the instrument and is completed at 600 m. An overlap correction function O(r) is used to reconstruct the backscatter signal over that region (Biavati et al., 2011). Narrow band interference filters, with 5 nm bandwidth, high transmission and negligible temperature dependence, select the backscattered light from the sky background. A cube polarizer is used to further divide the

radiation at 532 nm in the components parallel and perpendicular to the polarization of the emitted light. Additional polarizers are placed in front of the detectors, which are miniature photomultiplier modules (Hamamatsu 6780-20) with low thermal noise





(less than 10 counts/s at 25 °C). A characterization cross-talk between the channels has been performed following the method outlined in Snels et al. (2009), resulting to be negligible. The photomultiplier signals are amplified with a bandwidth of 250 MHz then recorded both in current and in photocounting mode by an acquisition card (Embedded Devices, APC-80250DSP).

The two records can be merged during data processing. In current mode, the photomultiplier signal is filtered through a 15 MHz low pass to avoid aliasing and then digitized into an 8 bits waveform, at adjustable sampling rates. The sample duration can be set to 12.5, 25, 50 or 100 ns and the waveform is reconstructed from 1024 samples, providing a spatial resolution from 1.875 to 15 m and a spatial range from 1.875 to 15 km. The system has been carefully checked for linearity throughout the measurement dynamical range. An absolute calibration of the channels gain ratio was also performed before the deployment.

In photon counting mode the photo impulses which are above an adjustable threshold are counted in 1024 consecutive time bins, whose length may be set from 25 to 1000 ns in 25 ns increments, providing a spatial resolution from 3.75m to 150 m and a spatial range from 3.75 to 150 km. The dead time estimated from the maximum photon counting rate result to be 6 ns, and its effects are taken into account following Donovan et al. (1993). In both modes the first 24 bins are collected before the laser shot and used for measuring background light. The acquisition card provides the sum of the signals integrated over a

user defined time interval that can range from 1 s to possibly several tens of hours. Data are stored in the memory board of the system. An external computer is used to access the system via USB or TCP/IP connection. A visualization of the measurements in real time is possible for system checking or for alignment. A synopsis of the system specifications is reported in table 1. In the Palau campaign configuration, the system was set to operate every night for 10 hrs, providing 5 minutes averaged vertical profiles of atmospheric elastic backscattering with a vertical resolution of 30 m, extending from the ground up to 30 km.

**2.2 Data processing**

In elastic lidars, the two physical quantities of interest, the particle backscattering $\beta_a(r)$ and extinction $\alpha_a(r)$ coefficients, must be determined from the elastic backscatter equation:

$$P(r) = \frac{EC}{r^2} O(r) \left[ \beta_a(r) + \beta_m(r) \right] \exp\left( -2 \int_0^r \left[ \alpha_a(s) + \alpha_m(s) \right] ds \right) \tag{1}$$

where $P(r)$ is the power of the backscattered radiation received by the lidar telescope from range $r$, $E$ is the transmitted

laser-pulse energy, $C$ is the lidar constant including its optical and detection characteristics, $O(r)$ is the overlap function and $\beta_m(r)$ and $\alpha_m(r)$ are the molecular backscatter and extinction coefficient respectively, that can be derived by meteorological data of air density and molecular scattering theory. To determine the two unknowns, the particle backscattering $\beta_a(r)$ and extinction $\alpha_a(r)$, a relationship must be assumed between them, often in the form of a constant extinction-to backscatter ratio, the so-called Lidar Ratio (LR). This parameter is an intensive aerosol property, strongly depending on its size, shape and

composition.

The backscatter ratio (BR), defined from the particle ($a$) and molecular ($m$) backscattering coefficients as:

$$BR = \frac{\beta_a + \beta_m}{\beta_m} \tag{2}$$


is then derived with the Klett inversion method (Klett, 1985) with LR assuming piecewise constant values in regions where clouds or different typologies of aerosols were present. To identify such regions, the values of BR, altitude and volume depo-

larization ratio $\delta$ are iteratively inspected during processing. The volume depolarization ratio $\delta$ is defined as:

$$\delta = \frac{\beta_a^{cross} + \beta_m^{cross}}{\beta_a^{par} + \beta_m^{par}} \qquad (3)$$

where *par* and *cross* refers to the backscattered light with polarization respectively parallel and orthogonal to that of the laser. These values are iteratively inspected during the data processing to recursively adjust the LR accordingly. For instance, when thin liquid or ice clouds are identified, LR there is set to values known from literature respectively to 19 sr (Chen et al.,

2002) and 29 sr (O'Connor et al., 2004). The LR for aerosol may easily range from 20 sr in the case of marine aerosol (Papagiannopoulos et al., 2016) to 80 sr for biomass burning aerosol (Weinzierl et al., 2011). We used a constant aerosol lidar ratio set to 26 sr where no clouds were present (Dawson et al., 2015). The values of $\beta_m$ were retrieved from temperature and pressure profiles (Collis and Russell, 1976) by co-located radiosoundings which were launched twice a day by the Weather Service Office of Palau, accessible through http://weather.uwyo.edu/upperair/sounding.html.

The calibration altitude for BR was chosen typically between 8 and 12 km when free of aerosol and clouds. We set there the calibration value of BR tot 1.02, as suggested in Kar et al. (2018). The uncertainty associated with the data from the lidar used in this study is extensively discussed in Cairo et al. (2012) and Rosati et al. (2016). At a given altitude, it depends on the attenuation of the signal from the ground, which is affected by low level clouds and aerosol. For the present purposes, a typical upper limit to the absolute random errors on BR and $\delta$ in the tropospheric range may be quantified to be 0.1 and 5%

respectively.

An independent evaluation of the cirrus extinction, and hence of the cirrus LR, has been implemented following the approach of Young (1995), obtaining the value of the cloud transmittance determined from the elastically scattered lidar signals from clear regions below and above the cloud. When this approach does not produce results, due to optical thickness too small or noise of the profile below and/or above the cloud, a fixed value of LR=29 sr was assumed (Chen et al., 2002).

Multiple scattering should be considered when cirrus clouds are analyzed. The effect depends on the lidar FOV and on the optical thickness of the clouds, increasing when the two get larger and becoming not negligible when $\tau$ approaches unity. It tends to produce observed extinctions and depolarization respectively smaller and greater than the real (effective) ones; the effect on the backscattering coefficient tends to be less important (Bissonnette, 2005). Different correction algorithms have been proposed although there is still no consensus on a univocal and rigorous correction method. We have followed the procedure

suggested in Chen et al. (2002), multiplying our $\tau$ and the retrieved LR by a multiple scattering factor $\eta$ given by:

$$\eta = \frac{\tau}{e^\tau - 1} \qquad (4)$$

In our analysis, this correction ranges from close to 1 in very thin clouds to 0.7 for the thickest ones. No corrections were made to the backscattering and depolarization coefficients. The inversion of the lidar data delivers profiles of Backscatter Ratio BR and hence of particle backscatter coefficient $\beta_a$, and volume depolarization $\delta$. From the latter, a value of the particle

depolarization $\delta_a$ can be obtained following Cairo et al. (1999). The uncertainty associated to $\delta_a$ very much depends on the





associated BR and in our data, can be as large as 100% for a single measurement on very thin clouds. However this uncertainty is the result of random errors that do not produce a bias in the mean values of the measurements, so that the averages presented in our work may not be precise but are accurate.

## 2.3 Data analysis

Threshold values of 1.15 for BR were used to identify the cloud base $z_{bottom}$ and top $z_{top}$ as limits for the calculation of the cloud integrated characteristics; clouds were identified as such in an altitude interval in which condition R > 1.15 was continuously met for at least 150 m. For them, the vertical extension of the cloud $\Delta z$ and optical thickness $\tau$ have been computed. The mid cloud optical altitude $z_o$, average temperature, average particle depolarization, average potential temperature have been computed as weighted averages over the cloud vertical extension, with the weight given by the cloud particle backscatter

coefficient $\beta_a$. The mid cloud geometrical altitude $z_g$ is defined as the average of the cloud top and base altitude.

A backtrajectory analysis was conducted to investigate the thermal and convective history of the cloudy airmasses. The convective origin of the air masses was analysed making use of the TRACZILLA Lagrangian model (Pisso and Legras, 2008), a variation of FLEXPART (Stohl et al., 2005) that interpolates vertical velocities and heating rate to the position of the parcel from the hybrid grid using log pressure or potential temperature. Each parcel simulation releases a cluster of 1000 back-

trajectories, representative of a generic aerosol tracer. The release points are computed from the cirrus cloud position from the lidar measurements with a time step of 3 hr along the time axis and a vertical sampling of 5 points equidistant in pressure between the top and the bottom of the cloud. The trajectories are reconstructed back in time for 30 days in a geographical domain covering the whole globe. The meteorological fields at $1°\text{x}1°$ resolution are taken from the ERA-Interim ECMWF reanalysis at 3hr resolution, using diabatic vertical motion. The convective influence is individuated from the 3-hourly brightness

temperature (BT) images at 11 $\mu$m from the climate quality Gridded Satellite dataset GRIDSAT-B1 (GRIDSAT78 BI) (Knapp et al., 2011). A convective source is therefore individuated when, in a specific geographical bin, a trajectory is traveling below (has a temperature warmer than) a convective cloud top level, selected from the satellite measurement with BT < 230 K, as similarly done in Tzella and Legras (2011) and Tissier and Legras (2016). More details on the trajectory-convective clouds coupling methods can be found in Bucci et al. (2020) and Legras and Bucci (2019). The age of theair masses is computed as

the time intercurred between the time of release of the cluster, i.e. at the time of the lidar observation of the airmass, and the convective cloud crossing.

## 3 Measurements

### 3.1 Meteorological context

Palau is located on the north western edge of the TWP. Its average daily temperature is 28 ° C throughout the year with very

small changes from season to season. Its weather is influenced by the meandering of the Intertropical Convergence Zone, extending across the Pacific just north of the equator, most intense in the Northern Hemisphere wet season. The main wet





season is from May to October, affected by West Pacific Monsoon that brings heavy rainfall, while the driest season is from February to April (Kubota et al., 2005). Winds come from the north-east from December to March, then revert to westerly between May and July, to September and December.

The lidar system started operating on the 16 of February 2016 and stopped on the 25 March. That coincided with a whole period of the Madden-Julian Oscillation (MJO), the eastward moving disturbance of clouds, rainfall, winds, and pressure that traverses the tropics in 30 to 60 days on average. The upper panel of figure 1 reports the MJO phase diagram during the 50 days of the campaign. We remind that such diagrams represent the magnitude of the first two empirical orthogonal functions of the combined fields of near-equatorially averaged 850-hPa zonal wind, 200-hPa zonal wind, and satellite-observed outgoing

longwave radiation (OLR) data (Wheeler and Hendon, 2004). Points representing sequential days are joined by a line, and their distance from the origin is related to the strength of MJO cycle. Their position with respect to the eight quadrants into which the diagram is divided indicates the geographical region in which the MJO phase brings an increase in convection. Labels S and E on the diagram marks the days of beginning and end of our field campaign. According to that, we should have expected an enhanced convection in the second half of the campaign.

Clouds were observed throughout the campaign, with a prevalence in the second part as depicted in the lower panel of figure 2 , where a time serie of clouds observation is reported; each point represent the average of BR within clouds on a 3 hours time window. Colour codes the same average for the particle depolarization.

Figure 2 reports the average vertical temperature profile (left panel) during the campaign time frame. On average, the Cold Point Tropopause (CPT) temperature is 192 K at 17400 m at 382 K potential temperature on average; that altitude is nearly

coincident with the Lapse Rate Tropopause (LRT) , defined as the level at which the lapse rate becomes less than 2 K/km and the average lapse rate is less than 2 K/km for 2 km above this level, The Level of Neutral Buoyancy (LNB), which we consider to be the one at which the potential temperature equals the equivalent potential temperature at the ground, is at 11600 m at 348K potential temperature on average. In our case it can be shown that this level coincides with the Level of Minimum Stability (LMS), where the vertical gradient of the potential temperature attains a minimum, and thus can be taken as the lower

boundary of the TTL (Sunilkumar et al., 2017).

On the left panel of the figure the time series of temperature anomaly profiles observed during the campaign are reported. The altitudes of the CPT (blue dots) are also displayed. The pronounced wave-fronts in the stratosphere that are seen to descend from the stratosphere and sometimes penetrating below the CPT are the result of large scale tropical wave activity. We can see here how the structure and variability of the TTL are greatly affected by these fluctuations induced by such gravity waves or

Kelvin waves, that influence tropopause height, temperature, cloud occurrence, (Fueglistaler et al., 2009).

Prevalent winds were from East, veering South with altitude and becoming southerly close to the Tropopause.

## 3.2 Clouds vertical distribution and morphology

Figure 3 reports the distribution of BR > 1.15 observations with respect to altitude. Each observation is a 5-min average over an altitude interval of 30 m. Low level clouds top extends up to 2.5 km, then a relatively clear altitude range is observed. High





altitude clouds, which are the ones we will focus our attention on, appear from 10 km upward, with two peaks of maximum occurrence respectively at 12 km and at 15 km; then their occurrence tapers off until the tropopause is reached.

Figure 4 reports the statistics of cloud optical thickness versus geometrical mid cloud altitude. Here and in the figures that follow, the reported data are from 5 min averages of lidar vertical profiles with 30 m vertical resolution, and a cloud is defined as an altitude interval not thinner than 150 m where the condition BR>1.15 is continuously met. The continuous horizontal

lines indicate the $\tau$ value that separates altostratus and thin cirrus from subvisible cirrus ($\tau < 0.03$). Most of our observations are SVC, mainly collected between 15 km and the Tropopause; the distribution of the SVC has a second maximum in the lower part of the TTL, between 11 and 13 km altitude. The thin cirrus clouds are distributed more evenly in altitudes up to 15 km, with a slight tendency to get thicker with increasing altitude; no significantly thick cirrus are present at the Tropopause. In figure 5 the statistics of $\tau$ vs the geometrical cloud thickness $\Delta z$ are shown. In spite of the wide spread of the data, an almost

linear increase of $\tau$ with $\Delta z$ is apparent, with two different slopes respectively for the SVC and the thin cirrus classes. Figure 6 reports the relative difference between the geometrical and optical mid cloud altitude ($z_g$-$z_o$)/$z_g$, vs altitude, colour coded with $\tau$. Positive values indicate that the backscattering is more intense in the upper part of the cloud, which is then radiatively more effective than the lower part. This parameter may vary depending on the distribution of Ice Water Content (IWC) inside the cloud, in turn affected by the microphysics of particle formation and redistribution by sedimentation and evaporation. In

our case, the lower and upper parts of the cirrus appear to produce the same scattering effect for small to medium values of $\tau$, indicative of an even distribution of backscatter inside, which we may take as a proxy for the distribution of IWC. Conversely, the thickest clouds tend to have lower backscatter in their bottom part with respect to the top part, with few exceptions for the highest ones; this is arguably due to mass redistribution by sedimentation and subsequent evaporation.

## 3.3 Clouds optical properties

The trend of depolarization with the altitude, i.e. with decreasing temperature, shows a compact linear relationship with a progressive increase towards higher altitudes, as shown in figure 7. This linearity is apparent from 10 km to slightly below the tropopause. It is interesting to note that approaching the tropopause, a different behavious sets in, so that between 15 and 17 km an entire range of depolarization values is also observed. It is possible to show that the data associated with the compact linear increase of depolarization toward higher altitude are associated with particle backscatter coefficients $\beta_a$ covering the entire

range of their variability (in our data, between $10^{-8}$ and $10^{-4}$ m$^{-1}$ sr$^{-1}$). Conversely, those depolarization data at high altitude which are almost evenly distributed between 10 and 60%, are associated only with medium to low values of $\beta_a$. The fact that different optical typologies of clouds coexist in the upper part of the TTL can be drawn also from the inspection of the relation between average particle depolarization $\delta_a$ and optical thickness $\tau$, shown in figure 8. There, two branches can be discerned: for medium to high values of $\tau$ associated to thin cirrus, those arguably topping at 15 km in figure 4, there is a decreasing

trend of $\delta_a$ vs $\tau$, while for small to medium values of $\tau$ associated to SVC, the particle depolarization is more spread through its variability range. The analysis of the LR adds another piece of information, as it is a parameter linked to the average size dimension of the particles: the lower the LR, the larger the particles. In figure 9 we report the occurrence of LR vs $\tau$. For thin clouds LR stays around the value of 29 sr, often measured and reported in literature while for optical thickness typical of SVC,





LR tends to get to higher values, suggesting the presence of smaller particles in the clouds. A similar lesson can be learned
by looking at the histogram of occurrence of LR and depolarization, shown in figure 10. There, the LR values around 29 sr,
typical of thin cirrus, are associated with depolarization significantly higher than those associated with LR around 40 sr, which
in the previous figure were linked to optical thickness typical of SVCs. A similar spread in LR has been reported elsewhere: for
instance, from one year of cirrus obsservation in the Amazonian basin, Gouveia et al. (2017) report how the most frequent LR
were between 18 and 28 sr, peaking at 25 sr for opaque cirrus and at about 21 sr for thin cirrus, and showing a bimodality for
SVC with two peaks respectively at 15 sr and at 44 sr. Our observations of clouds with LR around 40 sr are found mainly in the
highest part of the TTL at colder temperatures, as shown in figure 11 where LR is reported in relation to the cloud temperature.
So in the upper part of the TTL, two classes of clouds are present, namely thin cirrus with high depolarization and low LR, and
SVC with medium to low depolarization and larger LR. The optical parameters that we used to discern these two cirrus classes
are indicative of differences in the average shape and dimension of the particles that compose them.

### 3.4 Clouds close to the tropopause

It is worthwhile zooming in on what happens close to the Tropopause, therefore we move to a reference system centered at the
altitude of the CPT, and we replot the statistics of the particle depolarization data with respect to the distance from the CPT in
figure 12; again we are able to see the presence of two modes in the depolarization, pariculary evident from 2 to 1 km below
the tropopause, and around the tropopause, one with high values and one with medium to low values of the depolarization.
It is worth noting that clouds, especially those with medium to low depolarization values, do not extend significantly above
the CPT. In figure 13 for the same subset of observations depicted in figure 12, we report the statistics of the simultaneous
occurrence of particle depolarization and local temperature anomalies, these latter calculated as deviations of the temperature
profile measured by the concomitant radiosounding with respect to the average temperature profile during the campaign, as
reported in the right panel of figure 2. It is evident that cloud occurrence is associated with cold temperature anomalies. It is
noteworthy that clouds with high depolarization are present throughout the whole variability range of cold anomalies, while
those with medium to low depolarization occur only when the temperature drops 2 to 3 K below the average. It can be shown
that such correlation between cloud occurrence and negative temperature anomaly is not present for clouds at lower levels.

## 4 Trajectory analysis

The trajectory analysis allows possible linking of the optical characteristics of the clouds with the thermal and convective
history of the airmass. Clusters of 1000 trajectories were launched backward from the altitude and time of the clouds obser-
vations, and the averaged trajectory over each cluster, together with its dispersion, was used for further analysis. Along each
of the averaged backtrajectories, the minimum temperature encountered $T_{min}$ and the time elapsed since that, $t(T_{min})$, was
computed, together with the derivatives of temperature $dT/dt$, potential temperature $d\theta/dt$ and pressure $dp/dt$ at the time of
lidar measurements, computed as averages over the past 18 hours. Moreover, the following quantities were also computed: the
time during which the air mass remained below the temperature attained at the lidar observation $t(T(t) < T_0)$; the minimum



potential temperature $\theta_{min}$ and the maximum pressure $p_{max}$ and the relative times elapsed from the lidar observation $t(\theta_{min})$ and $t(p_{min})$.

For each averaged backtrajectory, the 'convective fraction' - defined as the percentage of the trajectories in the cluster that had met convection - was also computed. The time elapsed since the most probable convection encounter was defined as the time from the maximum increase of the convective fraction in the cluster.

This information obtained from the analysis of the retro-trajectory was connected to the measured optical parameters: depolarization, optical thickness and LR. We give here a brief resume of the main results; a more extensive analysis and supporting figures are available in the supplementary material.

Most of the clouds in the upper part of the TTL are stationary or experience a slight warming, while some of those in the lower part, especially among the thickest ones, are cooling (figures S4, S5 and S6 in Supplementary material).

Most of the cloud observations had max temperature differences along the backtrajectories which are below 5 K, with few exceptions for clouds in the mid TTL between 12 and 15 km, with medium to high optical thickness and medium to low depolarization, which experienced temperature differences as large as a few tens of Kelvin (figures S1, S2 and S3 in Supplementary material).

Most of the clouds encountered their minimum temperature $T_{min}$ less than 48 hrs before observations therefore relatively recently, with few exceptions of clouds below 14 km, with high optical thickness and medium to low depolarization for which the temporal distance from the minimum of temperature along the backtrajectory was as large as 100-200 hours (figures S13, S14 and S15 in Supplementary material).

Generally, the clouds spent less than 48 hrs below the temperature $T_0$, so indicative of relatively recent origin, with the exception of some clouds present both in the upper part of the TTL, and below 14 km, which spent several days below $T_0$ (figures S16, S17 and S18 in Supplementary material).

A more interesting picture came from the analysis of the maximum potential temperature difference along the trajectory, indicative of the altitude jump: while for clouds below 16 km this difference is smaller than 10 K, above that level $\theta_{max} - \theta_{min}$ is consistently greater, ranging from 15 to 25 K (figure S19, S20 and S21 in Supplementary material). In fact, above that altitude the potential temperature gradient begins to assume stratospheric characteristics. Moreover, the clouds in the TTL and particularly in its upper part had met $\theta_{min}$ several days before the observation, while on the contrary, in the backtrajectories from clouds at lower levels $\theta_{min}$ was temporally significantly closer to the observations, going back to some hours up to one or two days before (figures S22, S23 and S24 in Supplementary material).

The convective influence on the cloud airmass spans between 20 and 60 % , evenly spread across altitude, somewhat larger for the highest and optically thinnest clouds (figures S28, S29 and S30 in Supplementary material); the time elapsed from the most likely convective encounter is very often below two days, except for very few cases, scattered throughout the altitude range, when it is greater (figure S31, S32 and S33 in Supplementary material).

It is worthwhile noting that, for the set of high level SVC with low optical thickness and depolarization and high LR which may be depicted on the left side of figures 9 and 10, they have the highest potential temperature difference along the backtrajectory (figures S19, S20 and S21 in Supplementary material), and the longest time interval from $\theta_{min}$ (figures S22, S23





and S24 in Supplementary material) and below $T_0$ (figures S16, S17 and S18 in Supplementary material), possibly suggesting an origin other than the recent convection one.

## 5  Discussion

The analysis shows how clouds occurrence extends throughout the whole TTL, with two peaks, one at 10-13 km and the other at or slightly below the CPT; clouds do not extend significantly above it. It is noteworthy how only those upper TTL clouds are largely affected by local atmospheric temperature anomalies.

Depolarization increases with height and generally decrease with temperature. In the upper part of the TTL, depolarization may vary over a wide range, with some of the highest and thinnest SVC clouds attaining medium to low depolarization and LR greater than those found in other SVC at the same altitude, and in thicker clouds below. In situ measurements, reviewed in Lawson et al. (2019), have shown that particle shapes in fresh anvil outflows are markedly different from those in in situ formed cirrus, the latter showing a prevalence of bullet rosettes and polycrystals, virtually absent in the convective cirrus dominated by single crystals and aggregates of crystals: plates, double plates, columns, and irregulars, with some needles, stellars, and dendrites. This difference reflects different formation mechanisms, as ice may form in convective updrafts prior to reaching the homogeneous freezing level (-38° C). Cirrus from aged outflows have intermediate characteristics. Moreover, in the upper part of the TTL where in situ formation is likely predominant, measurements show in SVC the prevalence of small quasi-spheroids and plates, in concentrations of 1-10 cm$^{-3}$, while in the lower part of TTL where cirrus can more likely be generated by convection, particles are bigger and attain lower concentrations.

Some of the observed SVC may have had rather long life times, and are associated with negligible lifting in the hours preceding the observation. The study of the convective influence is not conclusive, so we may speculate that this class of SVC may have originated by in situ formation, with nucleation processes that induce morphologies, sizes and concentration of ice crystals that differs from those that we can observe at lower altitudes (11-14 km), where the depolarization and the optical thickness is higher and the LR is lower. Such in situ formation may be triggered by the activity of tropical waves that may induce temperature anomalies as large as 2-3 K close to the CPT, while the clouds in the lower part of the TTL, more likely originated by convective events, are dependent on local temperature conditions only for their subsistence and not for their formation.

## 6  Conclusions

Five weeks of night lidar observations of of equatorial cirrus clouds have been presented and their morphology and optical and geometrical characteristics have been discussed, also in the light of a trajectory analysis aiming at connecting their properties with the thermal and convective history. The lidar-derived depolarization shows a linear increase with altitude up to the top of the troposphere, where a full range of depolarization values is present. For a subset of these high altitude clouds, low depolarization and optical thickness values are associated with high values of LR, indicative of small particles. The absence



of recent significative uplifting in the histories of those airmass suggests a mechanism of formation by in situ condensation, triggered by temperature fluctuation. The fact that the presence of small particles in cirrus particle size distribution increase in percentage with decreasing temperature i.e increasing altitude, and with increasing time since convective influence, has been reported by Woods et al. (2018).

Conversely, the majority of cirrus in the TTL have thermal and convective histories compatible with formation from convective outflows, bringing up moisture and ice. In the thickest of them, the negative difference between the optical and geometrical mid cloud altitude gives a hint that larger ice crystals are being removed from the TTL by sedimentation, leaving smaller particles on optically thinner cirrus that can survive for several hours. Compared to observations in other climatically important regions, observations of equatorial cirrus in the TWP are scarce so this work, albeit limited to lidar observations during a relatively short time frame, contributes to enriching observations. The work would have benefitted from the simultaneous use of balloon hygrometers to better explore the effect of cirrus clouds on the distribution of water vapor in the TTL. An observational campaign with balloon hygrometers and the deployment of the more performing lidar COMCAL by the Alfred Wegener Institute will help to better define the morphology and microphysical processes in TTL in this region.

*Data availability.* Lidar data are temporarily available at https://stratoclim.icg.kfa-juelich.de/AfcMain/CampaignDataBase/DataMicro, they will migrate to a definitive database in the future. Contact the first author for further information.

*Author contributions.* FC, MDM, MS, LDL, set up and extensively tested the system and software; FC, MDM deployed and maintained the system in campaign; FC and AS performed the data analysis; SB, BL, AK, provided bactrajectories and convective analysis; SG did a preliminary study on backtrajectory during his master thesis; FC wrote the manuscript which was reviewed by everyone; the software for analysis was produced by FC, MS, LDL, AS, SB.

*Competing interests.* The authors declare that they have no conflict of interest.

*Acknowledgements.* This study is funded by the StratoClim project by the European Union Seventh Framework Programme under grant agreement no. 603557. The authors gratefully aknowledge the support of Justus Notholt, Institute of Environmental Physics, University of Bremen and Katrin Müller, Alfred Wegener Institute, Potsdam that organized and guided the activities of the Atmospheric Observatory in Palau; Sharon Patris, Coral Reef Research Foundation and Patrick Tellei, Palau Community College, Palau, and Ingo Beninga and Wilfried Ruhe, Impres GmbH, Bremen, which very much assisted us remotely and on the field; finally Maurizio Viterbini, ISAC-CNR now retired without whose invaluable technical support this work would not have been possible.



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

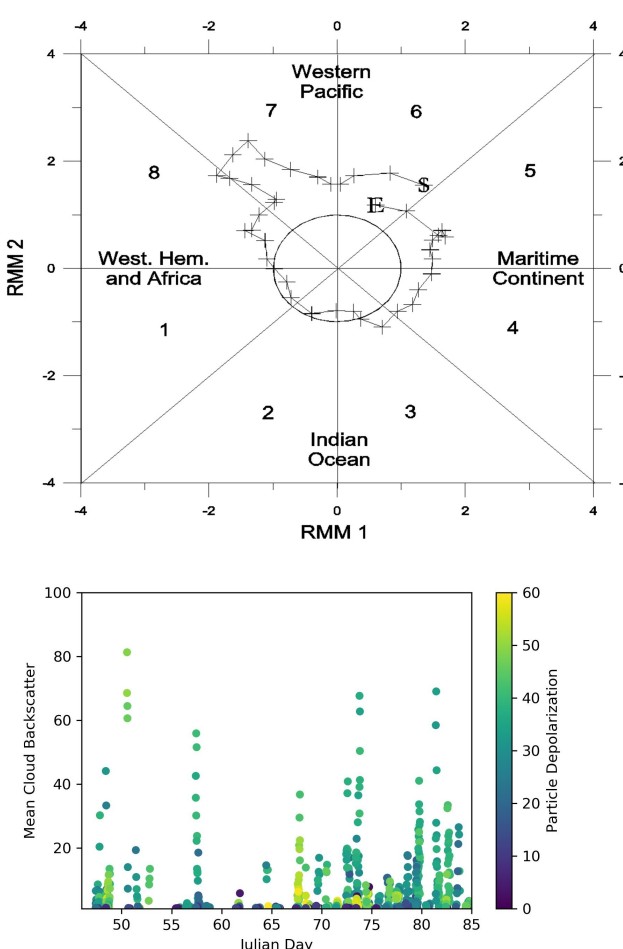

**Figure 1.** Upper panel: Madden-Julian Oscillation phase diagram for the campaign duration. Days of start and ending of the campaign are marked respectively with an S and an E. Data from the Australian Geovernment Bureau of Meteorology (http://www.bom.gov.au/climate/mjo/). Lower panel: Timeseries of observations of clouds Backscatter Ratio (BR) during the campaign. The colour codes the Particle Depolarization. Data points are averages over 300m of the lidar profile and over 3 hrs of observations. Observations with BR<1.15 have not been reported.

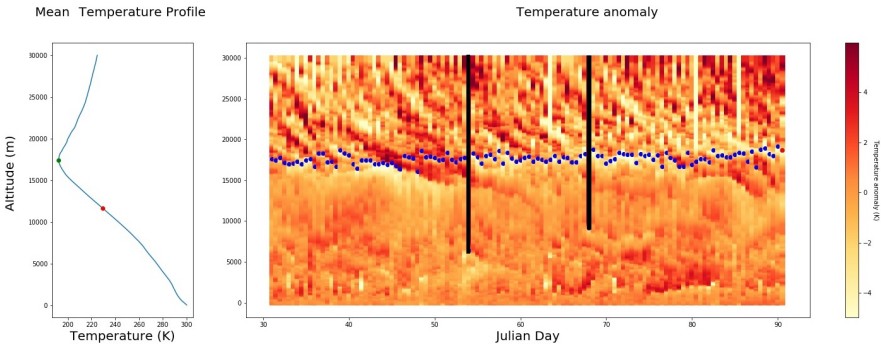

**Figure 2.** Left panel: Average temperature profile during the campaign timeframe. The green dot marks the Cold Point Tropopause (CPT) at 17400m, the red dot marks the Level of Neutral Buoyancy at 11600 m. The Tropical Tropopause Layer (TTL) can be considered to be withint these two limits. Right panel: Temperature anomaly with respect to the average temperature profile on the left. Data are from 12-hrs radiosoundings routinely launched by the Weather Service Office of Palau from Koror, Palau (station 91408) downloaded from: http://weather.uwyo.edu/upperair/sounding.html





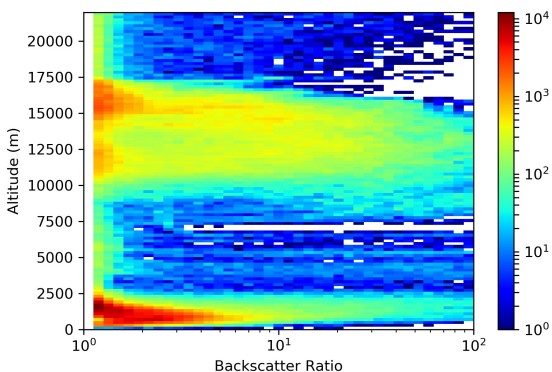

**Figure 3.** Distribution of Backscatter Ratio observatons vs altitude. Data are 5 min averages of lidar vertical profiles, with 30 m vertical resolution. The colour codes the number of samples in each bin. Only data with BR>1.15 have bee reported.



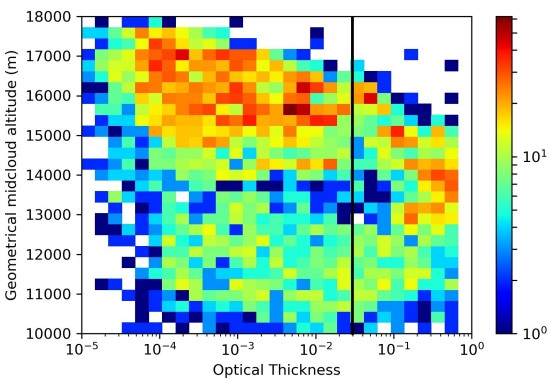

**Figure 4.** Distribution of cloud optical thickness observations vs mid cloud altitude. The colour codes the number of samples in each bin. The thick black line reports the optical thickness threshold value for SVC.





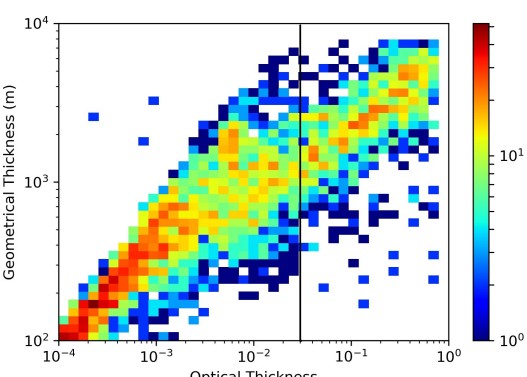

**Figure 5.** Distribution of cloud optical vs geometrical thickness. The thick black line reports the optical thickness threshold value for SVC. The colour codes the number of samples in each bin.

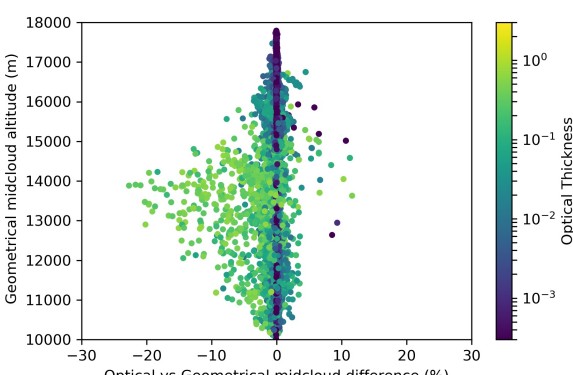

**Figure 6.** Scatterplot of the relative difference between the optical and geometrical mdcloud altitude, vs the geometrical mid cloud altitude, colour coded with the cloud optical thickness.




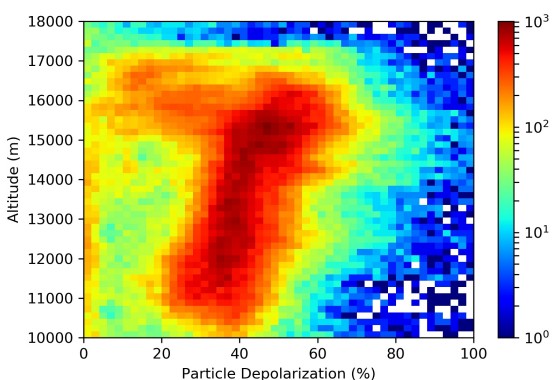

**Figure 7.** Distribution of particle depolarization vs altitude. Only datapoints with BR>1.15 are reported.





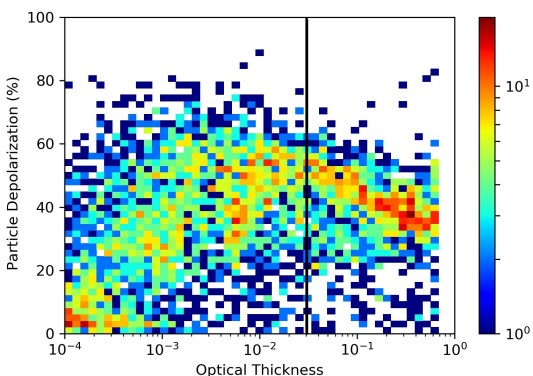

**Figure 8.** Distribution of particle depolarization vs optical thickness. The thick black line reports the optical thickness threshold value for SVC.



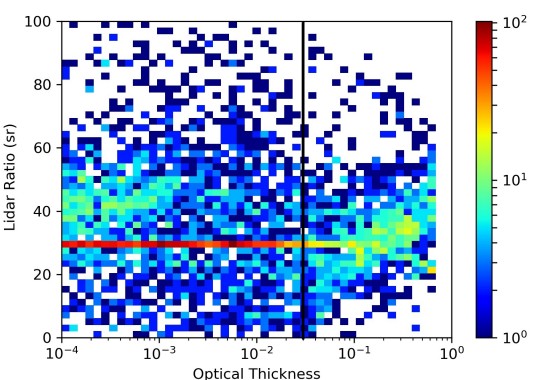

**Figure 9.** Distribution of Lidar Ratio (LR) vs optical thickness. The thick black line reports the optical thickness threshold value for SVC. The observations that accumulate along the line at LR=29 sr are those for which the inversion according to Young (1995) did not produce convergence to a result for LR. For these observation, LR was set at 29 sr by default.



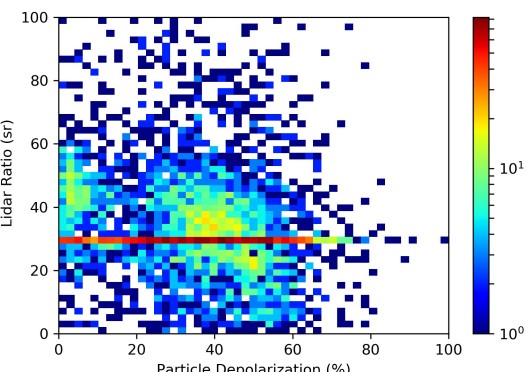

**Figure 10.** Distribution of Lidar Ratio (LR) vs Particle Depolarization. The observations that accumulate along the line at LR=29 sr are those for which the inversion according to Young (1995) did not produce convergence to a result for LR. For these observation, LR was set at 29 sr by default.



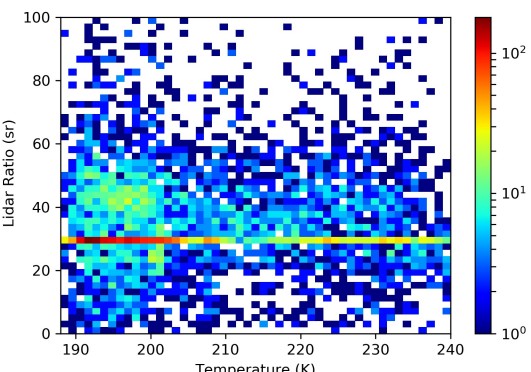

**Figure 11.** Distribution of Lidar Ratio (LR) vs mid cloud Temperature. The observations that accumulate along the line at LR=29 sr are those for which the inversion according to Young (1995) did not produce convergence to a result for LR. For these observation, LR was set at 29 sr by default.





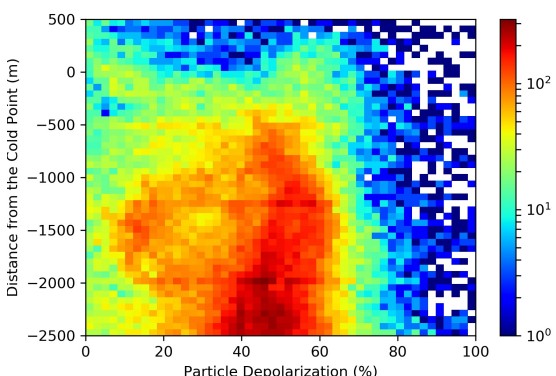

**Figure 12.** Distribution of particle depolarization observations vs distance from the Cold Point Tropopause (CPT).





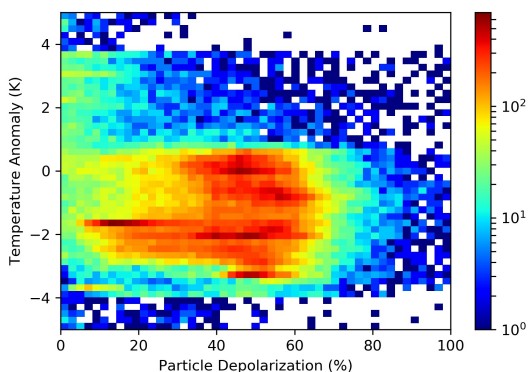

**Figure 13.** Distribution of particle depolarization observations vs temperature anomaly with respect to the average temperature profile as reported in the left panel of figure 2. Only data from clouds within 2500 m from the CPT are reported.





**Table 1.** Lidar specifications

| Technical specifications of the lidar system | |
| --- | --- |
| Detected wavelengths | 532 nm (two polarizations) |
| Laser type | Nd-YAG (SHG 532 nm) |
| Pulse duration | 1 ns |
| Laser repetition rate | 1 kHz |
| Laser output energy | 1 mJ at 532 nm |
| Telescope diameter | 20 cm |
| Telescope type | F/1.5 Newtonian |
| Telescope field of view | 0.7 mrad |
| Beam divergence | 0.4 mrad, full angle |
| Filter bandwidth | 2 nm |
| Vertical resolution | From 3.75 to 150m in photoncounting mode |
| | From 1.875 to 15m in current mode |
| Vertical range | 1024×Vertical resolution |
| Time resolution | down to 1s |

t