# Peer review of "Lidar observations of Cirrus clouds at Palau island ( $7^{\circ}33'$ N, $134^{\circ}48'$ E)"

_Atmospheric Chemistry and Physics, 2020_

## Referee Comment (RC1) · Anonymous Referee #1 · 25 Nov 2020

The authors present a study of cirrus morphology and optical properties (depolarization, optical thickness and LR), based on a short term measuring period. For the identified cirrus layers the cloud optical properties for the 532-nm wavelength were derived and an empirical multiple-scattering correction was applied, based on the optical depth values. Authors also present a trajectory analysis giving some possible links between the optical characteristics of clouds with the thermal and convective history of the air mass. The main results of the study are of interest. However, the methodology part needs further work and improvements, as even the cloud boundary detection is not clearly described. The paper is suitable for publication in ACP. However, I recommend a major revision of the manuscript, after considering some general and specific issues detailed below in my review.

General comment

According to authors, the detection of the cirrus clouds is made after applying the following threshold: Threshold values of 1.15 for BR. With only this threshold, are authors confident that they exclude from the analysis any transported elevated layer? Moreover, have you checked the SNR, before applying the cirrus detection?Using a standard BR value, can be applicable to cases that no systematic errors occurred (e.g wrong background subtraction). However, the SNR (signal to noise ratio) should also checked.

How can authors explain the detection of cirrus clouds with depolarization values less than 10%? Figures 7, 8, 10, 12, 13 are reporting cirrus depol values starting from 0%? Have you checked the SNR of these causes? The depolarization values are really surprising for cirrus clouds. Moreover, the integration time of 5-min average, could have restrict the accuracy of the depolarization ratio.

Concerning the multiple scattering correction, I would suggest the authors to make clear that the derived optical depth, may contain significant biases due to the multiple scattering, corrected by Chen et al., (2002), with eta depending on the optical depth of the cloud layer. The authors, should justify better, the reason for adopting this approach. Moreover, the authors claim that (Page 6 , Line 177) "It tends to produce observed extinctions and depolarization respectively smaller and greater than the real (effective) ones". However, few lines latter, they state that "No corrections were made to the backscattering and depolarization coefficients."  How can you deal with that? Is this the reason for the very low depol values presented? The authors should provide more details and explanation.

Authors claim that "When this approach does not produce results, due to optical thickness too small or noise of the profile below and/or above the cloud, a fixed value of LR=29 sr was assumed (Chen et al., 2002)". The authors should provide more details about the errors introducing in their statistics with this choice.

The authors claim that LR is indicative of small particles. However, they should be more careful, as this parameter depends also on the particle orientation relative to the laser beam.

The structure of section 3 separates the Clouds vertical distribution and morphology (#3.2) and Clouds optical properties (#3.3). However as it is written and structured the text introduces optical properties (optical depth values) in the first part (#3.2). I propose a reconstruction of these Sessions or a change on the title of the second session.

Terms should be clearly defined the first time they appear in the manuscript (e.g. Line 4 UTLS). Also, replace the abbreviations in the Conclusion part.

Please provide more references in the Discussion part.

Specific comments

You are referring to the measuring period either from 15 February to 25 March or from 16 February to 25 March. Please correct.

Page 5 line 129. "An absolute calibration of the channels gain ratio was also performed before the deployment". Authors must provide details for the procedure.

Page 6, Line 179. The authors claim that "Different correction algorithms have been proposed although there is still no consensus on a univocal and rigorous correction method". Please check and provide as references the model of Eloranta, 1998 and Hogan, 2006.

*Eloranta, E.: Practical model for the calculation of multiply scattered lidar returns, Appl. Optics, 37, 2464–2472, https://doi.org/10.1364/ao.37.002464, 1998.*
*Hogan, R. J.: Fast approximate calculation of multiply scattered lidar returns, Appl. Optics, 45, 5984–5992, 2006.*

Figures 2b and 6. The color scale used makes the figures hard to read and to distinct values, especially for values close to zero. The authors should consider choosing a different color map.

Page 12, Line 357. "Depolarization increases with height and generally decrease with temperature". Please provide references to enhance this statement.

Figure 5. Why cirrus clouds with geometrical thickness less than 150m are plotted in the Figure? According to line 254, page 9, thinner clouds than 150m are excluded from the analysis.

Figure 2. The size of the axis labels should be improved to be readable.

Figure 3. Replace "Distribution of Backscatter Ratio observatons vs altitude. Data are 5 min averages of lidar vertical profiles, with 30 m vertical resolution. The colour codes the number of samples in each bin. Only data with BR>1.15 have bee reported" with "Distribution of Backscatter Ratio observations vs altitude. Data are 5 min averages of lidar vertical profiles, with 30 m vertical resolution. The colour codes the number of samples in each bin. Only data with BR>1.15 have been reported".

Page 12, Line 360. Double "in" written.

Page 10, Line 288. Replace "obsservation" with observation

Page 3, Line 84. Replace airmasses with air masses

Page 7, Line 191. Replace "R" with "BR"

Page 7, Line 196. Replace "airmasses" with "air masses"

Page 7, Line 209. Replace "theair" with "the air"

Page 8, Line 236. Replace "," with "."

Page 9, Line 272. Replace "behavious" with "behaviour"

Page 10, Line 310. Replace "airmass" with "air mass"

Page 12, Line 376. double "of"

Page 12, Line 360.

---

## Referee Comment (RC2) · Anonymous Referee #2 · 25 Nov 2020

The paper of Cairo et al. presents one month of lidar observations of the characteristics of cirrus clouds in one of the most important tropical regions for troposphere – stratosphere exchange. As available atmosphere data from that area is sparse, this paper is a very valuable contribution to our knowledge about clouds in that region. So, the paper is scientifically significant and it is also well suited for publication within ACP. The scientific as well as the presentation quality of the paper is very good, therefore I recommend a publication after minor revisions.

In general, the paper can be improved in two ways. Not all of the results and figures of the lidar data analysis need to be in the main text, but some could rather be moved to the supplementary material. This specifically holds for figures 6, 9, 10, 11. The result of fig. 6 is related to the Ice Water Content (IWC), but without having such data

available, the discussion related to fig. 6 is a bit speculative. In a similar manner, figs 9 – 11 can be discussed in a much shorter and more concise way and the figures moved to the supplement. The paper cannot give no solid links between Lidar Ration (LR) and cloud properties, last not least as the LR is not directly measured by this instrument. Therefore, I recommend to move this part to the supplement, too.

The other suggested improvement to the paper is a discussion about the relevance of this observational data set with respect to clouds above the tropics in general. It is great that the observational time spans a whole Madden-Julian cycle. However, it is still just one month of a particular year, so any more general conclusions should be related (and limited?) to this. One possibility would be referring the Feb.-March 2016 observations to longer records of satellite observations. This aspect should be covered at least in the discussion part of the paper and would help to corroborate the conclusions.

The supplementary material as compiled at the moment is mainly a collection of plots of point clouds, which rarely allow to draw substantial conclusions. It shows the high amount of work put into the analyses, but I do not find the results enlightening, as they mainly reflect a very high variability and little correlations. The authors need to present (explain in the supplement) what can be learnt from these plots, otherwise they are not helpful and could be part of a dedicated data paper.

Some detailed suggestions for improvements as follow:

Abstract, line 13: "SVC" needs to be defined here

Introduction, line 83-84: this sentence is not well written: "with a Lagrangian trajectory analysis of tropospheric airmasses entering the stratosphere." What do you want to express?

Instrument, line 122: should read: A characterization OF cross-talk . . .

Line 166: "We set here . . . BR to 1.02" not "tot"

Line 176: first appearance of tau, needs to be defined here, not in line 192

Line 191: "BR > 1.15" not "R > 1.15"

Line 209 "theair" is missing a space

Measurements: Line 224: which wind data is used here? Is this from local radio soundings or from a reanalysis model? Please specify.

Line 254: Figure 4 shows one vertical line not several horizontal lines.

Line 298: "particularly" instead of "particularly"

Conclusions, line 377: "of of"

Figures: In general, many figures have a quite small lettering on the axes' labels, these most likely need to be enlarged before publication. This is particularly necessary for fig. 2 axes labels.

Fig. 3 caption: "bee" Fig. 6 caption: "mdcloud"

I look forward to a revised publication of this paper!

---

## Author Comment (AC1) · 22 Feb 2021

**Response to Reviewer 1**

We thank Reviewer 1 for the careful assessment of the manuscript and valuable suggestions.

The data have been reprocessed to take into account the remarks made in his/her review.

The main changes have been:

i.     The data are now averaged over 10 minutes instead of 5 (line 253 now changed to "… data are from 10 min averages…"

ii.    The BR threshold value to detect cirrus clouds (above 10 km) clouds has been increased from 1.15 to 1.2; moreover, an additional threshold on the SNR has been used, so that a cirrus cloud is now identified as (line 254 now changed: "… a cloud is defined as an altitude interval not thinner than 150 m where the condition BR>1.2, and SNR lower than 0.5 on the parallel channel is continuously met."

This reprocessing has brought some changes in the overall results of the study:

1.   the number of data points with particle depolarization below 10% has been greatly reduced. Compare new fig. 7 to the old one, and note the virtual disappearance of data points with aerosol depolarization <10%, now basically restricted to a layer between 16 and 17 km. Depolarization values around 10% however still exist; they are associated with the lowest values of BR. We acknowledge that if low BR values resulted overestimated during an inaccurate calibration process, they can lead to an underestimation of the particle depolarization. We have no reason to suspect such inaccuracy in the determination of the BR at high altitude, however the possibility exists and – given the BR and volume depolarization values at play – an overestimation of 0.5 in the BR may induce a relative overestimation as high as 50% in the particle depolarization. This cannot be ruled out and we have stated it explicitly in line 276 that now reads (… low values of beta_a. In particular, particle depolarization values as low as 10%, which are atypically low for cirrus clouds, have been observed in association with lowest values of beta_a. These low values can be relatively more affected by an inaccurate signal calibration process which may in turn induce inaccuracies in the determination of particle depolarization. We have estimated such inaccuracies to be no greater than 50% on the particle depolarization value."

[Figure]

Fig.1: Left panel old fig. 7 in the manuscript, displaying right panel, new fig. 7.

2.   A reduction of the number of clouds with low optical thickness. This is evident in new figure 4,  the bidimensional (Optical Thickness, altitude) Probability Distribution Function. The reduction is more marked in the 11000-12000 m range, less in the 15000-17000 m range. However, the shape of the PDF is maintained. Similarly, in figure 5 (geometrical thickness, Optical Thickness), the peak of the PDF has shifted from very low thickness values to values close to the SVC threshold, maintaining

the overall shape of the function.

[Figure]

Fig.2 Left panel old fig 4 in the manuscript, displaying right panel nef figure 4.

3. The greatest change was in the LR values. The LR values greater than 29, previously reported mainly at high altitudes and low temperatures, which were associated with low depolarization and low values of optical thicknesses, are no longer so apparent (see new figures 9,10,11). This forces us to drop the discussion from line 281 to 294 which have been replaced with:" The analysis of the LR obtained with the Young procedure shows that in the majority of cases LR is distributed between 20 and 40 sr, with a peak around 30 sr and without showing particular dependencies on the mean depolarization, temperature or optical thickness of the cloud (see figs. S2, S3 and S4 in Supplementary material).". Moreover from line 348 onward there is no reference to LR and now reads : "It is worthwhile noting that the high level SVC with low optical thickness and depolarization have the highest potential temperature difference along…"

The old figures 9 ,10 and 11 have been moved to the supplement material as suggested by referee2.

[Figure]

Concerning the multiple scattering (MS) correction, line 179 has been modified as: "We have followed the procedure suggested in Chen et al. (2002) that assumes that the real $\tau$ values can be obtained by multiplying the observed ones by a factor $\eta$ depending on the $\tau$ of the cloud layer itself. In our case, $\eta$ was calculated iteratively by applying the correction to the observed $\tau$ multiple times, until the consistency between the real and observed $\tau$ and $\eta$ was achieved. In our analysis, the $\eta$ correction ranges from close to 1 in very thin clouds to 0.58 for the thickest ones, which is however a small portion of our data.  This latter value can be taken as the order of magnitude of the possible bias on the largest optical depths due to multiple scattering effects."

We have not considered this effect on the depolarization. The MS effect should depend, among other factors, on the penetration depth inside the cloud. So, if MS plays a role on depolarization, observed depolarization should increase with increasing penetration within the cloud. We have checked if there was a systematic increase in depolarization as penetration into the cloud increased and we did not find any;

rather the depolarization remains constant even in the thickest clouds, throughout their thickness. So we thought we could overlook the effects of MS on depolarization. We have stated it in the new manuscript, where line 182 now reads "No corrections were made to the backscattering and depolarization coefficients. In fact, the effect of multiple scattering in depolarization is to increase the observed depolarization as the penetration of the lidar pulse into the cloud increases. We inspected the cloud depolarization profiles and found no systematic increase with altitude within the cloud. We therefore considered the effect of multiple scattering in our depolarization, and ever more so on backscattering, to be negligible."

Concerning the LR retrieved with the Young technique and its effect on the data statistics, it is clear that the uncertainty of the lidar ratio is the single most important source of inaccuracies in elastic lidar retrievals.

We have changed line 174 to: "… a fixed value of LR=29 sr was assumed (Chen et al., 2002). The choice of LR has an effect on backscattering and extinction retrievals. The distribution of our retrieved LR is almost evenly disperse around the fixed value LR=29, ranging from 20 to 40 sr-1.  If we consider such variability as an estimation of the LR uncertainty, and given the size of the optical thickness involved which is often low, the retrieval of the backscatter coefficient may be considered accurate and so for the particle depolarization, while the extinction, and hence the retrieved optical depths, are more affected as the relationship between the two and LR is linear, can be inaccurate up to a factor of 2."

Sections 3.2 and 3.3 have been merged. All acronyms have been checked and reported explicitly on first use. The discussion was deepened and new references were added. See reply to Reviewer 2.

The Discussion part has been greatly reshaped, and  more references have been added, namely: Massie et al., 2010; Sassen et al., 2008; Sassen et al, 2009; Nazaryan et al., 2008; Virts and Wallace, 2010; Vitrs and Wallace , 2014; Zou et al., 2020; Luo and Rossow 2004, Wang et al., 2020; Sunilkumar et al., 2005.

Specific Comments

1. 15 February, corrected.
2. Line 129 now reads: "An absolute calibration of the channels gain ratio was also performed before the deployment, following the procedure outlined in Snels et al. (2009)."
3. Different correction algorithms have been
4. Line 179 has been modified as: "… proposed (see for instance Eloranta et al. 1998, Hogan et al. 2006) although corrections or adaptations of single scattering retrieval algorithms to take into account multiple scattering effects are not straightforward." We ae not experts of these topic, we had the impression of no univocal consensus on a particular correction method, reading the Bissonnette survey: "…  models of multiple scattering but the main inputs to drive these models are actually the medium properties we wish to correct for. Iterations to derive true values from "effective" values are possible but there is almost always a missing input not available from the retrieval algorithms under study, e.g., the phase function." so that "work is continuing at a steady pace to devise practical retrieval algorithms to account for or even exploit multiple scattering".
5. Figure 2b and 6 have been improved
6. Line 357 now reads: "In our dataset depolarization increases with height and generally decrease with temperature, as has been reported in other observations (Wang et al., 2020; Sunilkumar et al., 2005)"
7. cirrus clouds with geometrical thickness less than 150m which were apparent in fig. 5 for a software error, are now not present.
8. The readability of the labels has been improved.
9. Figure 3 captions has been corrected
10. and following. All typos have been corrected.

---

## Author Comment (AC2) · 22 Feb 2021

**Response to Reviewer 2**

We thank Reviewer 2 for the careful assessment of the manuscript and valuable suggestions.

Please note that, in the reviewed manuscript, the data have been reprocessed to take into account the remarks made by the Reviewer 1. For the sake of completeness, I report here an extract of my response to Reviewer 1. Please refer to that for further explanation on the impact of the reprocessing on the data:

The main changes have been:

i. The data are now averaged over 10 minutes instead of 5 (line 253 now changed to "… data are from 10 min averages…"

ii. The BR threshold value to detect cirrus clouds (above 10 km) clouds has been increased from 1.15 to 1.2; moreover, an additional threshold on the SNR has been used, so that a cirrus cloud is now identified as (line 254 now changed: "… a cloud is defined as an altitude interval not thinner than 150 m where the condition BR>1.2, and SNR lower than 0.5 on the parallel channel is continuously met."

This reprocessing has brought some changes in the overall results of the study:

1. The number of data points with particle depolarization below 10% has been greatly reduced, and are now basically restricted to a layer between 16 and 17 km.
2. The number of clouds with very low optical thickness is reduced.
3. The LR values greater than 29, previously reported mainly at high altitudes and low temperatures, which were associated with low depolarization and low values of optical thicknesses, are no longer massively apparent.

Figure 6 has been deleted from the main text and moved to the Supplementary Material (now fig. S1), and the text from line 260 to the end of the paragraph 3.2 has been changed as follows: "The vertical distribution of backscattering inside the cloud was investigated (see fig. S1 in Supplementary Material for further details). In many cases, the lower and upper parts of the cirrus appear to produce the same scattering effect for small to medium values of tau, indicative of an even distribution of backscatter inside, which we may take as a proxy for the distribution of IWC. Conversely, the thickest clouds tend to have lower backscatter in their bottom part with respect to the top part, with few exceptions for the highest ones; this is arguably due to mass redistribution by sedimentation."

Please note that to meet the suggestion of Reviewer 1, paragraphs 3.2 and 3.3 are now merged.

According to the reviewer's suggestion, figs 9, 10, and 11 have also been deleted from the main text and moved to the Supplementary Material (now figs S2, S3, S4). As a result, the discussion from line 281 to 294 has been changed to: "The analysis of the LR obtained with the Young procedure shows that in the majority of cases LR is distributed between 20 and 40 sr, with a peak around 30 sr and without showing particular dependencies on the mean depolarization, temperature or optical thickness of the cloud (see figs. S2, S3 and S4 in Supplementary material)."

Similarly from line 348 onward there is no reference to LR and now it reads : "It is worthwhile noting that the high level SVC with low optical thickness and depolarization have the highest potential temperature difference along…"

Following the Reviewer's suggestion, all arguments using LR as support have been removed from the manuscript. In particular, in previous line 349, 359, 372, 381,

Paragraph 5. "Discussion" has been deeply restructured and expanded (please refer to the revised manuscript), with the intention of placing the work in the more general context of studies on tropical

clouds and their seasonality. Additional references to satellite studies have been quoted, namely Massie et al., 2010; Sassen et al., 2008; Sassen et al, 2009; Nazaryan et al., 2008; Virts and Wallace, 2010; Virts and Wallace , 2014; Zou et al., 2020; Luo and Rossow 2004, Wang et al., 2020; Sunilkumar et al., 2005. Every onsideration on the LR have been eliminated.

Supplementary Material has been deeply reshaped and reduced as suggested, now hosting only the previous figures 6,9,10,11, now removed from the manuscript.  The study on the dependence of optical parrameters on the backtrajectory analysis has now been summarized in the manuscript.

Detailed suggestions:

Line 13: acronym explained.

Line 83-84: modified as: "Kremser et al. (2009), by following Lagrangian trajectories from the Troposphere until  their entering the Stratosphere, have demonstrated that…"

Line 122: corrected.

Line 166: corrected.

Line 176: optical Thickness tau defined for the first time in line 29.

Line 209: corrected.

Line 224: A reference to the data source has been added in the text (line 223-225 now reads: "  … two empirical orthogonal function RMM1 and RMM2 […]. RMM1 and RMM2 data are from the Australian Bureau of Meteorology website http://www.bom.gov.au/climate/mjo/") and in the figure caption.

Line 254: corrected.

Line 298: corrected.

Line 377: corrected.

The fonts of all figures have been enlarged.

Fig 3 caption has been corrected, as well as old figure 6, now in the Supplementary Material as fig S1.

---

## Referee Report (RR1)

The authors present a study of cirrus morphology and optical properties (depolarization, optical thickness and LR), based on a short term measuring period. This is the second time I have reviewed this paper. Authors have addressed all of the remarks I made regarding their initial draft. Below find my remarks in the revised version, so as to be answered in a final draft.

General comment

========================================================

1) Please note that figure numbers are missing in the manuscript.

2) I see inconsistencies between figure 6 and the rest of figures with particle depolarization values plotted. In this figure a significant number of values less than 5% are plotted, while in the rest figures, these pretty low values are missing. Can the authors explain the reason for that?

3) Are these plotted depolarization values, averaged values? And why the colour codes for each figure of the manuscript are different?

4) Have the authors checked temperatures for the cases with the low values of depolarization?

5) Page 10, Line 292. What do the authors mean with this statement : "We have estimated such inaccuracies to be no greater than 50% on the particle depolarization value". I am really confused.

Specific comments

========================================================

Page 8, Line 247. Replace "2 ," with "2,"

Page 8, Line 251. Replace "Lapse Rate Tropopause (LRT) ," with "Lapse Rate Tropopause (LRT),"

---

## Author Response (AR2)

Answers to Referee report 1

We thank the Anonymous Referee 1 for his/her careful review and valuable suggestions.

General comments:

1) Due to mysterious reasons and my mild inability to use the "latexdiff" package, the compilation of the pdf of the marked-up manuscript crashed when text differences were found in captions. So we separately marked those differences in a dummy-proof word file, then pdf-zed it and merged it to the latex one, exclusively for the purpose of marking changes. Unfortunately, in that "homemade" document, figure numbers are missing from the manuscript text. We apologize for this inconvenience that now it seems I have resolved, although I still could not explain how.

2) The Authors thank the Reviewer for this remarks that allows us to explicitly clarify a possible misunderstanding. In fact, in the figure 6, as well as in the figures 8 and 9, the data reported are the experimental points, i.e. 10-min average over an altitude interval of 30 m. In fig. 7 instead, the reported values of depolarization are averages over the cloud extent.

   We don't seem to see inconsistencies between fig. 6, 8 and 9. It might be an impression due to colour scales, which we now have modified to improve the readability of the plots: where possible, the figures now shares the same colour scale, see answer 3).

3) As explained in point 2) above, in figs 6, 8 and 9 only experimental points are reported, i.e. 10-min average over an altitude interval of 30 m. In fig. 7 instead, the depolarization values are averages inside the cloud. We have tried to make clearer this point in the manuscript changing lines 294-295 to: "… between the cloud averaged value of the particle depolarization $\delta_a$ and optical thickness…" and fig. 7 caption as: "Distribution of cloud averaged values of particle depolarization vs optical thickness. The thick black line reports the optical thickness threshold value for SVC. ". For what concerns the colour codes, the colour is not normalized, i.e. the bin colour represents the actual number of data points in the bin. Since the total number of data points in figure 6, 8, 9 is different (as different are the y axes), the figure colours cannot be compared. However to improve readability, we have replotted fig.9, which in fact accommodates the same dataset of fig. 8, with the same colour scale (i.e. we corrected "Only data from clouds within 2500 m from the CPT are reported." to "Only data of clouds from 2500 m below to 500 m above the CPT are reported.", as now read the modified fig. 9 caption.) and the same number of bins. Similarly, figs. 4, 5 and 7 whose dataset has the same dimensions, have been homogenized in terms of bin numbers and color scales.
   In the histograms' captions, where missing, a sentence has been added: "The colours code the number of data points falling inside the bin."

4) Supplementary material now hosts a Depolarization vs Temperature plot as Figure 1, there commented as: "Figure 1 reports the trend of particle depolarization with decreasing temperature, It shows a compact linear relationship, with a progressive increase of particle depolarization from 40\% to 60\% as temperature gets colder. Noticeable, in the range 200-190K, the presence of low depolarizing clouds, a behavior that deviates from the main trend, with the probability of observing low values of depolarization which seems to increase as temperature decreases." In the Manuscript, line 283 now reads "… i.e. with decreasing temperature (see Figure 1 in Supplementary Material)…." Line 290 now read "particle depolarization values below 20% start to appear when temperature drops below 200 K and at temperatures around 190 K reach values as low as 10%, which are atypically low for cirrus clouds, have been observed in association with the lowest values $\beta_a$. Very low values of $\delta_a$ can be observed in presence of large oriented crystals in clouds, typically planar crystals with their main faces aligned horizontally, (Platt et al., 1977; Noel et al., 2005)." The

quotation of the possibility of low depolarization due to oriented crystals is now added, thanks to the editor's suggestion.

5) Page 10, Line 292 now read "We have estimated that these inaccuracies are no greater than the 50% of the reported value for the aerosol depolarization. This is an upper limit, as the inaccuracy uncertainty greatly reduces for aerosol depolarization values accompanied by high backscattering.".

Specific comments:

We have corrected the typos on page 8 line 247, line 251. Thank you.